# Metabolite Predictors of Breast and Colorectal Cancer Risk in the Women’s Health Initiative

**DOI:** 10.3390/metabo14080463

**Published:** 2024-08-20

**Authors:** Sandi L. Navarro, Brian D. Williamson, Ying Huang, G. A. Nagana Gowda, Daniel Raftery, Lesley F. Tinker, Cheng Zheng, Shirley A. A. Beresford, Hayley Purcell, Danijel Djukovic, Haiwei Gu, Howard D. Strickler, Fred K. Tabung, Ross L. Prentice, Marian L. Neuhouser, Johanna W. Lampe

**Affiliations:** 1Cancer Prevention Program, Division of Public Health Sciences, Fred Hutchinson Cancer Center, Seattle, WA 98109, USA; ltinker@fredhutch.org (L.F.T.); beresfrd@u.washington.edu (S.A.A.B.); rprentic@fredhutch.org (R.L.P.); mneuhous@fredhutch.org (M.L.N.); jlampe@fredhutch.org (J.W.L.); 2Biostatistics Division, Kaiser Permanente Washington Health Research Institute, Seattle, WA 98101, USA; 3Vaccine and Infectious Disease Division, Fred Hutchinson Cancer Center, Seattle, WA 98109, USA; yhuang@fredhutch.org; 4Department of Biostatistics, University of Washington, Seattle, WA 98195, USA; 5Biostatistics Program, Division of Public Health Sciences, Fred Hutchinson Cancer Center, Seattle, WA 98109, USA; 6Department of Anesthesiology and Pain Medicine, University of Washington, Seattle, WA 98195, USA; ngowda@uw.edu (G.A.N.G.); draftery@uw.edu (D.R.); purcellh@uw.edu (H.P.); djukovic@uw.edu (D.D.); 7Department of Biostatistics, University of Nebraska Medical Center, Omaha, NE 68198, USA; cheng.zheng@unmc.edu; 8Department of Epidemiology, University of Washington, Seattle, WA 98195, USA; 9Center for Metabolic and Vascular Biology, College of Health Solutions, Arizona State University, Phoenix, AZ 85004, USA; haiweigu@asu.edu; 10Department of Epidemiology and Population Health, Albert Einstein College of Medicine, Bronx, NY 10461, USA; howard.strickler@einsteinmed.edu; 11Department of Internal Medicine, Division of Medical Oncology, College of Medicine and Comprehensive Cancer Center, The Ohio State University, Columbus, OH 43210, USA; fred.tabung@osumc.edu

**Keywords:** breast cancer, colorectal cancer, metabolite predictors, dietary biomarkers, metabolomics

## Abstract

Metabolomics has been used extensively to capture the exposome. We investigated whether prospectively measured metabolites provided predictive power beyond well-established risk factors among 758 women with adjudicated cancers [*n* = 577 breast (BC) and *n* = 181 colorectal (CRC)] and *n* = 758 controls with available specimens (collected mean 7.2 years prior to diagnosis) in the Women’s Health Initiative Bone Mineral Density subcohort. Fasting samples were analyzed by LC-MS/MS and lipidomics in serum, plus GC-MS and NMR in 24 h urine. For feature selection, we applied LASSO regression and Super Learner algorithms. Prediction models were subsequently derived using logistic regression and Super Learner procedures, with performance assessed using cross-validation (CV). For BC, metabolites did not increase predictive performance over established risk factors (CV-AUCs~0.57). For CRC, prediction increased with the addition of metabolites (median CV-AUC across platforms increased from ~0.54 to ~0.60). Metabolites related to energy metabolism: adenosine, 2-hydroxyglutarate, *N*-acetyl-glycine, taurine, threonine, LPC (FA20:3), acetate, and glycerate; protein metabolism: histidine, leucic acid, isoleucine, *N*-acetyl-glutamate, allantoin, *N*-acetyl-neuraminate, hydroxyproline, and uracil; and dietary/microbial metabolites: myo-inositol, trimethylamine-*N*-oxide, and 7-methylguanine, consistently contributed to CRC prediction. Energy metabolism may play a key role in the development of CRC and may be evident prior to disease development.

## 1. Introduction

Breast cancer (BC) and colorectal cancer (CRC) are the first and third highest incident cancers in women in the US, respectively [1]. Substantial evidence outlined in the Third Expert Report of the World Cancer Research Fund (WCRF)/American Institute for Cancer Research (AICR) continuous update project supports the premise that dietary patterns and lifestyle factors significantly influence the risk of these cancers [2]. The Expert Report emphasizes the importance of maintaining a healthy weight, engaging in regular physical activity, adopting a diet high in fruits, vegetables, whole grains, and dietary fiber, and reducing intakes of red meat, animal fats, and refined carbohydrates [2,3,4,5]. Further, evidence suggests that even moderate alcohol consumption can contribute to an increased risk of post-menopausal BC and CRC [6].

Diet is a complex mixture of nutrients, bioactives, additives, and other components that can contribute to the risk of cancer [7]. Some chemicals, such as heterocyclic amines and polycyclic aromatic hydrocarbons formed when meat or fish are cooked at high temperatures, may be directly carcinogenic [8]. Other nutrients, such as saturated fat or added sugars, may be linked with cancer risk indirectly through alterations in various signaling pathways, such as insulin or inflammation [9,10,11]. Saturated fat may also contribute to excess caloric intake and weight gain [12], while foods rich in fermentable fiber may lead to beneficial gut microbial community structure [13,14]. These exposures along with phenotypic information can be captured with high-dimensional tools applied to blood or urine, such as metabolomics. Metabolomics is the comprehensive, qualitative, and quantitative study of the small molecules in an organism and includes both aqueous and lipid metabolites [15]. The metabolome reflects both endogenous processes, as well as diet and other environmental exposures. Thus, it provides a sensitive approach for testing and tracing the involvement of altered biological pathways and networks associated with chronic diseases, such as cancer. Although metabolomics has been used extensively to search for biomarkers of early cancer detection [16,17,18], metabolomic profiles are now being used as risk markers associated with environmental exposures [19,20].

In this study, our aims were to find potential prediagnostic serum and urine metabolite predictors of BC and CRC using multiple metabolomics platforms that provided predictive power above and beyond well-established risk factors within the Women’s Health Initiative (WHI) Bone Mineral Density (BMD) subcohort. Specifically, comparing several variable selection and prediction models, we assessed the competitive performance for BC and CRC prediction using metabolites compared to prediction models with only demographic, clinical, and lifestyle covariates, and assessed whether metabolites improved the prediction performance when added to these well-established risk factors. Also, comparing the results across variable selection and prediction approaches provides an evaluation of the robustness of the selected metabolites and their prediction performance. These analyses may provide novel metabolite–cancer associations and mechanisms, particularly for diet-related metabolites.

## 2. Materials and Methods

### 2.1. Women’s Health Initiative

The WHI recruited 161,808 post-menopausal women from 40 clinical centers nationwide between 1 October 1993 and 21 December 1998 [21]. All women were 50–79 years old when they were enrolled in at least one of three clinical trials (CT; *n* = 68,132) or an observational study (OS; *n* = 93,676). The three WHI CTs were a randomized controlled clinical trial of menopausal hormone therapy, of low-fat dietary modification, and of calcium/vitamin D supplementation. The WHI BMD subcohort included all participants at three clinical centers (Birmingham, AL; Pittsburgh, PA; and Tucson, AZ, with satellite in Phoenix, AZ) (*n* = 11,020) chosen to maximize racial and ethnic diversity. All women provided core questionnaires including medical history, reproductive history, family history, medication use, dietary intake, and personal habits [21].

### 2.2. Case and Control Selection

Cases and controls for this analysis were selected from the WHI BMD subcohort. The eligible sample was restricted to women who had sufficient serum (300 µL) and urine (550 µL) samples from the same time point, prior to and closest to BC or CRC case diagnosis date, and required to have no missing covariate data (*n* = 10,451). Clinical outcomes were reported biannually in the CT until 2005 through the trial periods, then annually, and annually in the OS. An initial report of invasive cancer during cohort follow-up was confirmed by a review of medical records and pathology reports by physician adjudicators.

The cases were defined as earliest incident invasive BC or CRC so that the biospecimen collection would be comparatively proximate. Each of the 758 case women was matched 1-to-1 to a control woman, disease free at the case occurrence follow-up time, based on age (within 2 years; Table 1), WHI enrollment date (within 2 months to control for follow-up duration), and self-identified race or ethnicity; the closest match was selected based on criteria to minimize an overall distance measure [22]. In total, 54% of the selected sample were in the OS, 34% in the dietary modification (DM) trial, and 12% in the hormone trials (HT) (but not in the DM trial). Our final population included *n* = 758 adjudicated cancers (577 invasive breast and 181 colorectal) and *n* = 758 controls.

### 2.3. Metabololite Profiling

#### 2.3.1. Measurement of Serum Metabolites

*Targeted LC-MS:* Serum samples were analyzed by targeted LC-MS/MS using liquid chromatography coupled to a Sciex Triple Quad 6500+ Triple Quadrupole mass spectrometer equipped with an ESI ionization source as described previously [23]. The instrument was attached to two Shimadzu UPLC pumps, and the pumps were connected to an auto-sampler in parallel so that chromatography separation could be performed using two analytical hydrophilic interaction liquid chromatography (HILIC) columns independently, one for positive ionization mode and the other for negative ionization mode. Identical columns (Waters XBridge BEH Amide XP) were used for both separations, and the samples were injected for each column separately. While one column was performing separation and MS data acquisition in ESI+ ionization mode, the other column was equilibrated and readied for analysis in ESI mode. The LC-MS system was controlled using AB Sciex Analyst 1.6.3 software. Serum metabolites were extracted using methanol in a 1:2 (*v*/*v*) ratio, dried, and reconstituted in HILIC solvent. MS data acquisition was performed in multiple reaction monitoring (MRM) mode. Measured MS peaks were integrated using AB Sciex MultiQuant 3.0.3 software. A total of 304 metabolites were targeted (see Appendix A), of which 150 were detected with less than 20% missing values. A total of 304 metabolites were targeted, of which 150 were detected with less than 20% missing values.

*Lipidomics:* Separately, serum lipid metabolites were measured using the Sciex QTRAP 5500 Lipidyzer platform including the SelexION differential mobility spectrometry (DMS) method [24,25,26]. Serum lipids were extracted using dichloromethane/methanol, dried under nitrogen, and the samples reconstituted in 100 μL of 10 mM ammonium acetate in dichlormethane:methanol (50:50). Lipids were analyzed in multiple reaction monitoring in both positive and negative ionization modes, with and without DMS. The method targeted lipids in 13 major lipid classes: cholesterol ester (CE), ceramides (CER), diacylglycerol (DAG), dihydroceramides (DCER), free fatty acids (FFA), hexosylceramides (HCER), lactosylceramide (LCER), lysophosphatidylcholine (LPC), lysophosphatidylethanolamine (LPE), phosphatidylcholine (PC), phosphatidylethanolamine (PE), sphingomyelin (SM), and triacylglycerol (TAG; see Appendix A). Absolute concentrations of lipids were obtained based on 54 isotope-labeled internal standards. A total of 1070 lipids were targeted, of which 687 lipids that had less than 20% missing values were measured.

#### 2.3.2. Measurements of Urine Metabolites

*NMR spectroscopy:* Metabolite profiles from 24 h urine samples were analyzed by NMR spectroscopy using a Bruker Avance III 800 MHz NMR spectrometer. Each sample (300 mL) was mixed with 300 mL phosphate buffer in D_2_O (pH = 7.4) containing an internal standard, 3-(trimethylsilyl)propionic acid-2,2,3,3-d4 sodium salt (TSP). Data were acquired at 298 K using a one-dimensional pulse sequence with suppression of the residual water signal using presaturation. Spectral width, time domain points, relaxation delay, and number of transients were 10,000 Hz, 32,768, 2 s, and 64, respectively. The raw data were Fourier transformed after zero filling by a factor of two and multiplied using an exponential window function with a line broadening of 0.5 Hz. The resulting spectra were phase and baseline corrected and referenced to the internal standard, TSP. Metabolite peaks were identified using databases and relative concentrations for 59 metabolites were obtained (Appendix A). None of the metabolites had missing values.

*GC-MS analysis:* Urine metabolites were also analyzed by untargeted gas chromatography–mass spectrometry (GC-MS) method using an Agilent 7890A/5875C instrument [27]. Urine samples were treated with urease and methoxime prior to derivatization of metabolites using MSTFA (*N*-Methyl-*N*-trimethylsilyltrifluoroacetamide) containing 1% TMCS (2,2,2-Trifluoro-*N*-methyl-*N*-(trimethylsilyl)-acetamide, chlorotrimethylsilane). We measured a total of 267 metabolites, 107 of which were identified (MSI level 1 or 2; Appendix A), with none having greater than 20% missing values. Overall, more than 1000 metabolites were identified from serum and urine samples using these four complementary analytical platforms.

### 2.4. Metabolite Quality Controls (QC)

Analysis protocols used multiple layers of QC samples as well as isotope-labeled or unlabeled internal standards to assess instrument stability/performance during the analysis and help with normalization and metabolite quantitation. Different types of QCs used included: (a) unblinded instrument QC samples (commercially obtained pooled human serum from Innovative Research, Inc. (Novi, MI, USA)) run every 10 samples and at the beginning and end of each batch of samples; (b) blinded, pooled study samples (5% for urine; 10% for serum) interspersed with the biological study samples (3 QCs/batch of 27 study samples), used to normalize batches of samples over the run; (c) 17 split-sample blinded duplicates of study samples also interspersed with study serum and urine samples, used to calculate reported median metabolite coefficient of variation (CV) values; (d) isotope-labeled internal standards for targeted analysis of aqueous metabolite (*n* = 33) and lipids (*n* = 54) in serum, which enabled absolute concentration determination and ensured evaluation of instrument stability and data quality; (e) internal standard, TSP, used to assess the spectral quality, calibrate spectra, and help with data normalization of urine NMR spectra; and (f) FAME (fatty acid methyl esters) of different fatty acid chain lengths for retention time indexing and myristic acid-d_27_ for help with metabolite identification and data normalization, respectively. Median CVs of blinded pooled study QC samples for the four different platforms (two for serum analysis and two for urine analysis) across the samples were 2.9% for global NMR from 24 h urine, 6.4% for targeted lipidomics, 20.7% for targeted LC-MS/MS, and 45.4% for global GC-MS.

### 2.5. Statistical Analysis

#### 2.5.1. Participant Data

From the originally collected participant data, we selected a base set of covariates (age, chronologic time of visit, and race or ethnicity) and all demographic, clinical, and lifestyle covariates that were adjusted for in Prentice et al. [28]. Some categorical variables—including race or ethnicity, education level, and income—were recoded as binary variables. The base set of variables listed above and other demographic, clinical, and lifestyle covariates considered are summarized in Table 1. We considered all identified metabolites from the four metabolomics platforms with less than 20% missing data. For each outcome, we used all cases corresponding to that outcome and all controls (i.e., all 758 controls were used in predicting both outcomes), which has been shown to improve prediction performance [29].

#### 2.5.2. Imputing Missing Data

While the base set of covariates were measured on all participants in this study, there were missing data in some of the demographic, clinical, and lifestyle variables [BMI < 1%, waist circumference < 1%, smoking history < 1%, energy expenditure 6%, and intake of alcohol, calcium, folate, and red and processed meat missing for fewer than 3% of participants]. We used multiple imputation via chained equations [30] to perform imputations. We ignored the outcome in all imputation models to simplify our procedure for assessing prediction performance described below (see, e.g., [31]). For metabolomics variables, those with more than 20% missing values were removed toward ensuring robust results. For the remaining variables, half of the minimum nonzero value was used to impute the values that were below detection limits. For each platform, we created one set of multiple imputed datasets consisting of only the metabolites and base set of variables, and a second set of multiple imputed datasets consisting of the metabolites and all risk factor variables (including the base set of variables). More detail on the imputation procedure is provided in the Appendix A.

After each platform-specific imputation step was complete, we further processed the data following a similar specification to Zheng et al. [32]. In particular, outliers were truncated to within three times the interquartile range of the first and third quartile. For LC-MS and GC-MS metabolites, we normalized the data within each imputation round and batch using local polynomial regression fitting (in the R package loess) with tuning parameter set to 0.75 among quality-control samples.

To minimize the effect of possible correlated variables on our results and to study the utility of different platforms, we first considered each measurement platform (NMR, LC-MS, GC-MS, and Lipidyzer) separately. Then, for each platform, we performed analyses based on metabolomics alone, and established risk factors + metabolomics, with the base set of covariates (age, chronologic time of visit, and race or ethnicity) always included in each analysis. In a sensitivity analysis, we pooled the metabolites from all platforms together.

#### 2.5.3. Algorithms Used for Selecting a Set of Metabolites

In all analyses, we adjusted for the base set of variables to account for the sampling design [33]. We report adjusted analyses using only the metabolites and using the metabolites plus other established risk factors. For a given platform and set of adjustment covariates, to evaluate the robustness of variable selection and prediction performance, we applied three algorithms to select a set of metabolites and covariates to use in the final risk-prediction algorithm, described below. The first algorithm performed no variable selection (i.e., allowed all metabolites and covariates into the final prediction algorithm). The second was lasso regression [34] implemented in the R package glmnet, with tuning parameters selected using ten-fold cross-validation. We forced the base set of covariates into all lasso models to ensure proper adjustment for these variables. We then selected all variables with a nonzero estimated coefficient.

The final procedure used the Super Learner [35] implemented in the R package SuperLearner [36]. The Super Learner is a particular implementation of stacking models [37]; in this algorithm, a library of candidate learners is fit to the data, and cross-validation is used to create the convex combination of these candidate learners that minimizes a cross-validated loss criterion. In these analyses, we used the non-negative log-likelihood loss function. The resulting convex combination has both finite-sample and asymptotic guarantees on its performance [35]. Our candidate library consisted of elastic net regression [38], boosted trees [39], and random forests [40]. The R implementations of these algorithms and the tuning parameters used are provided in Appendix A. To perform variable selection using the Super Learner, we first computed a variable importance measure for each candidate algorithm: an estimated coefficient for the elastic net and a decrease in Gini impurity for both trees and forests. We then ranked the variables from most to least important by the algorithm-specific metrics and combined the ranks using the convex weights of the Super Learner. We then selected variables with weighted rank (weights based on the Super Learner; see the Appendix A) in the top 20. This ensures that algorithms with high weight in the Super Learner ensemble—implying that the algorithm has favorable cross-validated performance—have a large influence in selecting variables.

#### 2.5.4. Assessing Prediction Performance

After selecting a set of metabolites and covariates, we addressed the performance of these variables in predicting either BC or CRC. We fit two final prediction algorithms for each platform. The first was a simple logistic regression. The second was the Super Learner, using the same approach as described above. This resulted in four procedures based on variable selection: variable selection with the lasso, followed by either logistic regression (denoted lasso + GLM below) or the Super Learner (denoted lasso + SL) for prediction; and variable selection with the Super Learner, followed by logistic regression (denoted SL + GLM) or the Super Learner (denoted SL + SL) for prediction. We compared these four approaches with two that did not use variable selection: the Super Learner with all variables (denoted SL) and the Super Learner with all variables that used a library of candidate learners augmented with variable selection algorithms [denoted SL (with screens)]. Further details on these procedures are provided in the Appendix A.

Assessing prediction performance of sets of selected variables was complicated by the fact that these variables were not determined a priori [41]. We used cross-validation to assess the performance of a combined procedure for variable selection and prediction using the selected variables, whereby the selected variables and prediction algorithm were determined on training data and prediction performance was evaluated on independent data. We repeated this cross-validated procedure 100 times for each platform and set of adjustment variables (base set of variables only or all risk factor variables). We measured prediction performance using the cross-validated area under the receiver operating characteristic curve (CV-AUC). Detail on this cross-validated procedure is provided in Appendix A.

#### 2.5.5. Final Selection of Metabolites

We obtained a final set of selected variables from each platform by applying the variable selection procedure to the full set of observations for each imputed dataset; our final set consisted of those metabolites and covariates that were selected in over 70% of the individual imputed datasets. For each platform and set of adjustment covariates, we then took the union of the sets resulting from the two variable selection procedures. Our final set of metabolites was a further union of the platform-specific selected sets, while the final set of adjustment covariates was the unique covariates selected in any of the platform-specific analyses (Appendix A).

#### 2.5.6. Post Hoc Sensitivity Analyses

Prior studies within WHI cohorts suggest an interaction between HT use and insulin such that associations between obesity-related measures, i.e., BMI, adipokines, levels of insulin, etc., and both BC and CRC were only observed among non-HT users [42,43]. It has been proposed that oral HT exposes the liver to a large dose of estrogen, leading to altered hepatic protein synthesis. Because HT could potentially alter metabolites associated with BC and CRC in our analysis, we conducted a sensitivity analysis for both cancer outcomes, excluding women randomized to the active arms of the HT or who reported current HT use at baseline.

## 3. Results

Characteristics of the WHI BMD participants stratified by BC and CRC cases and controls are given in Table 1. The mean time between blood draw and cancer diagnosis was 7.2 years (IQR 2.4–11.6 years).

Of the four metabolomics platforms, the greatest prediction potential was observed with LC-MS, which targeted water-soluble metabolites in serum. We present the cross-validated performance of each procedure for predicting BC and CRC using the LC-MS platform in Figure 1. In the left panel, we see that the base set of covariates, forced into all prediction models, and demographic, clinical, and lifestyle variables alone were moderately predictive of BC, and similar across all six prediction procedures, with a maximum CV-AUC of 0.572 from the SL (with screens) procedure. Performance for predicting CRC based on addition of risk covariates was similar (absolute difference 0.014), at a maximum CV-AUC of 0.558. In the right panel, we overlay the prediction performance using the metabolites and the prediction performance using all covariates and the metabolites. Metabolites alone were not good predictors of BC (CV-AUCs at or below 0.5), and the prediction performance of risk covariates plus metabolites was similar to that of the covariates alone, without performance improvement. In contrast, for CRC, prediction performance was improved for all six algorithms when using metabolites alone or metabolites plus risk covariates, with a maximum CV-AUC of 0.593 based on metabolites alone and 0.608 combining metabolites and clinical variables from the SL algorithm with no variable selection.

Results for the remaining platforms tended to also be consistent across the six algorithms. GC-MS and Lipidyzer-detected metabolites provided little to no additional prediction performance for either BC or CRC over the clinical variables. NMR-detected metabolites tended not to increase prediction performance for BC over the clinical variables; for prediction of CRC, these metabolites had a performance comparable to the risk covariates and also led to a slight increase in prediction performance when added to the risk covariates. The full set of results are presented in Appendix A.

In Table 2, we present the selected risk covariates and metabolites for predicting both BC and CRC, as well as the estimated proportion of variation explained (PEV) by each metabolite. Several risk covariates shown to be predictive of BC (including Gail 5-year risk score) or CRC (at least one colonoscopy or colon polyp removed) were selected, lending validation of our selection results. For individual metabolites, the PEVs were similar and in the range of 0.21 to 0.25, suggesting that many of the metabolites do not differentiate prediction performance alone, but can result in differential prediction performance together. Glycerate explained the most variability (PEV = 0.25) in CRC (adjusted for all risk factor variables). Metabolites selected for CRC, along with function, are given in Table 3.

To assess the effect of performing variable selection and estimating prediction performance based on each platform separately, we performed a sensitivity analysis. In this analysis, we pooled the metabolites from each platform together after imputation but before the variable selection and prediction performance analysis. Here, we only fit the lasso + GLM algorithm since we observed similar performance across procedures in the primary analysis. Estimated prediction performance based on the pooled set of metabolites was similar to that observed for the lasso + GLM algorithm for the LC-MS metabolites: CV-AUCs of 0.554 (BC) and 0.58 (CRC). In Table 4, we present the set of selected risk covariates and metabolites, along with the estimated PEV. Many metabolites selected from this sensitivity analysis were also selected in the platform-specific analyses, more so for CRC than BC, which had few metabolites selected in the pooled analysis. The estimated PEV was also similar for most metabolites, with glycerate, which was positively associated with CRC, again providing the largest PEV.

## 4. Discussion

In this well-characterized cohort of post-menopausal women, we evaluated whether the addition of serum and urine metabolites from multiple platforms were equivalent to or provided improved prediction of BC and CRC, beyond well-established risk factors. For BC, risk covariates alone provided moderate predictive power, in the range of CV-AUC 0.57, with metabolites contributing no improvement. In fact, the highest CV-AUC using both risk covariates and metabolites was <0.56. Conversely, for CRC, the addition of metabolites, particularly serum aqueous species from the LC-MS platform, modestly improved prediction performance over risk covariates alone, from CV-AUC of 0.54 to 0.61. This improvement was consistent across various prediction algorithms and metabolite platforms and held whether we performed variable selection within each platform separately or after pooling all metabolites together.

While metabolites did not provide additional prediction power for BC in our analyses, of those that were selected, a large proportion were lipids (22 of 43 named metabolites) or metabolites related to lipid metabolism. Associations between lipids and BC align with accumulating evidence associating excess adiposity, especially after menopause, with increased BC risk [2,5,44]. Obesity is associated with systemic inflammation, insulin resistance, altered steroid metabolism, and other metabolic derangements—factors mechanistically linked to carcinogenesis [44,45,46]. However, few lipids were selected in sensitivity analyses where all metabolites were pooled across all platforms. Moreover, variables, such as alcohol intake, waist circumference and BMI, current estrogen use, and Gail 5-year risk score, were superior to metabolites in predicting BC.

In contrast, mainly aqueous and urinary metabolites were selected in predictive models for CRC, with few if any lipids. Twenty-two different named metabolites were selected in the various prediction algorithms that contributed to CRC prediction, the majority consistently selected across procedures. Several were related to energy metabolism, including adenosine, 2-hydroxyglutarate, and glycerate, with additional metabolites, *N*-acetyl-glycine, taurine, threonine, and lysophosphatidyl choline [LPC (FA20:3)] related to fatty acid metabolism in particular. These metabolites suggest altered metabolism, a hallmark of cancer. An even larger proportion of metabolites were involved in protein metabolism. Histidine, *N*-acetyl-glutamate, and allantoin were inversely associated with CRC. As has been previously reported in two other large prospective cohorts, higher circulating histidine, even up to 10 years prior to diagnosis, was associated with reduced risk of CRC [47]. *N*-acetyl-glutamate functions as a cofactor in ureagenesis, converting nitrogen from protein to urea acids such as allantoin [48]. Other amino acids and peptides were positively associated with CRC, potentially reflecting higher protein intakes. For example, trimethylamine *N*-oxide (TMAO) is elevated in blood after consumption of fish or foods rich in choline and carnitine, such as red meat, eggs, and dairy products, which can be converted to trimethylamine by gut microbes [49], and subsequently to TMAO by hepatic enzymes in the liver. This metabolite has also been previously linked with CRC [50,51]. The branched-chain amino acids isoleucine and leucic acid, a metabolite of leucine, hydroxyproline, methylguanine, and n-acetyl-neuraminate, are all animal protein derived metabolites. Higher intakes of animal protein, especially red and processed meat, are a known risk factor for CRC [52,53]. In addition to generation of ATP, adenosine, along with the purine 7-methylguanine and pyrimidine uracil, are involved in DNA and RNA synthesis as well as participating as signaling molecules. Lastly, myo-inositol is a biomarker found in whole grains. These metabolites as a group are highly representative of dietary exposures and support conclusions from the WCRF/AICR Third Expert Report indicating probable or convincing evidence for several dietary components contributing to CRC risk, i.e., red and process meat, heme-containing foods in general, and low intake of fruits and non-starchy vegetables, but less so for BC risk, with strong evidence limited to alcohol intake [2].

While metabolite biomarkers have historically been used for cancer detection, studies are now using pre-diagnostic metabolites to examine environmental exposure and cancer risk. To date, 10 studies have focused on BC, with varying metabolite signatures [19,54,55,56,57,58,59,60,61,62,63]. These studies, including information on population, sample size, follow-up time, and analytic platforms have been extensively detailed in His et al. [19]. Most studies report significant associations with one or more metabolites, most commonly specific lipid species and amino acids. However as has been noted, except for steroids, there is little metabolite overlap across studies, including our own, making it very difficult to conduct comparisons [19,60].

Similar work in large prospective cohorts using pre-diagnostic samples has been conducted in the context of CRC. A case-control study nested within two Shanghai cohorts identified several serum phosphatyidylcholines and phosphatidylethanolamines that were inversely associated with CRC, suggesting that dysregulation of glycerophospholipids may contribute to CRC [64]. In the Prostate, Lung, Colorectal, and Ovarian Cancer Screening Trial, an inverse association was reported between leucyl-leucine, a metabolite representing incomplete protein catabolism, and CRC risk after eight years of follow-up, although the association did not remain significant after adjusting for multiple comparisons [65]. In the European Prospective Investigation into Cancer and Nutrition, concentrations of two lipid species—hydroxysphingomyelin C22:2 and acylakyl-phosphatidylcholine C34:3—were significantly inversely associated with CRC risk using a targeted metabolomics approach [66], with nine additional features, including two potentially annotated ceramides, reported in a follow-up analysis using untargeted lipidomics [67]. Investigators also identified a metabolite signature of greater body size, i.e., BMI, waist circumference, and waist-to-hip ratio, associated with a CRC. These metabolites were mainly related to amino acids and lipids, some of which were reversible with weight loss in a small subset of participants in a weight-loss pilot intervention [68]. In a multicenter study, a panel of 17 urine metabolites separated CRC patients from controls, with two providing good prediction in post hoc analyses (AUC of 0.86): diacetylspermine and kynurenine [69]. Another untargeted metabolomics approach was employed in the Cancer Prevention Study II Nutrition Cohort, where six named metabolites were related to CRC risk, including guanidinoacetate, 2′-I-methylcytidien, vanillylmandelate, bilirubin (E,E), *N*-palmitoylglycine, and 3-methylxanthine [70]. Finally, using untargeted metabolomics on plasma obtained up to 26 years prior to diagnosis in the Northern Sweden Health and Disease Study, seven features were found to be associated with CRC risk, two of which were identified as pyroglutamic acid, an amino acid derivative, and hydroxytigecyline, an antibiotic metabolite [71]. In both of the latter two studies, efforts to replicate previous findings in prospective cohorts were unsuccessful, except for 3-hydroxybutyric acid [71]. While the majority of metabolites selected in our predictive analyses for CRC were also novel, histidine, TMAO, and hydroxy proline were previously reported in other studies [47,50,51,69].

The strengths of this study include a well-characterized cohort, the novel use of four different metabolomics platforms, inclusion of both pre-diagnostic serum and urine, and several variable selection and prediction algorithms, which all yielded similar results, lending confidence to our findings. Further, our results are comparable to those reported by Wang et al. [72], using an alternate statistical approach in this population [73]. Our study population comprised post-menopausal women and may not be generalizable to other populations. We did not adjust for medication use, which may alter metabolite concentrations [74]; however, sensitivity analyses excluding women randomized to HT or women reporting use of HT at baseline yielded similar prediction performance results for both cancer outcomes. Other limitations include those commonly associated with metabolomic studies. A single data point may not be sufficient to adequately capture environmental or dietary exposures, and different metabolites measured may represent either exogenous exposures or alterations in endogenous processes. Further, our detected metabolite coverage of all pathways is incomplete. Nonetheless, we identified a panel of metabolites that were associated with risk of CRC and are biologically plausible.

In summary, we report a panel of metabolites associated with CRC risk in a subset derived from a large prospective cohort of post-menopausal women. That we identified more pre-diagnostic metabolites predictive of CRC than BC may reflect a more established, comprehensive set of metabolites available for CRC. Even for CRC, however, it is worth noting that the predictive contributions of metabolites were modest relative to established risk factors alone, and the results are largely disparate across this and other studies. This may be due, in part, to differences in populations, analytic platforms, length of biospecimen storage times, and statistical approaches. Further studies with repeated sampling, pooled analyses, and expanded metabolite platforms in diverse populations are needed to strengthen the comparison of results across studies.

## Figures and Tables

**Figure 1 metabolites-14-00463-f001:**
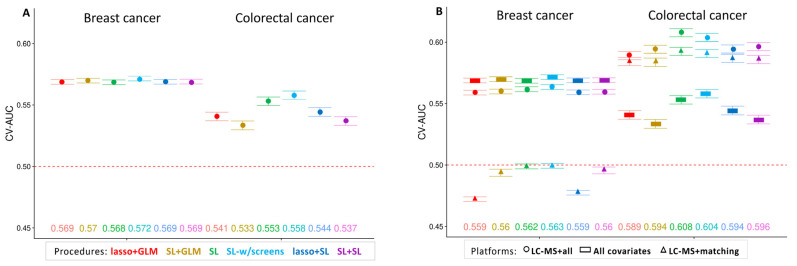
Cross-validated area under the receiver operating characteristic curve (CV-AUC) averaged over 100 Monte Carlo replications of each variable selection + regression procedure for predicting breast cancer and colorectal cancer, with 95% confidence intervals (CIs). (panel **A**) an analysis using only the covariates. (panel **B**) analyses using covariates only (circles), LC-MS metabolites + base set of covariates (triangles), and LC-MS metabolites + all covariates (squares). Point estimates of CV-AUC are provided at the bottom of each panel (on the right-hand panel, the point estimates correspond to the LC-MS metabolites + all covariates analysis).

**Table 1 metabolites-14-00463-t001:** Demographic, clinical, and lifestyle characteristics of the breast (BC) and colorectal cancer (CRC) cases and controls in the WHI Bone Mineral Density subcohort ^1^.

Characteristics	BC Cases (*n* = 577)	CRC Cases (*n* = 181)	Controls (*n* = 758)
*Demographic factors*			
Age (yrs)	62; [56, 68]	64; [58, 69]	63; [57, 68]
Body Mass Index (kg/m)^2^	27.73; [24.45, 32.45] *	28.05; [24.9, 31.98] *	27.16; [24.19, 31.18]
Waist circumference (cm)	86; [77, 96] *	87; [78, 99] *	84; [76, 93] *
Self-reported race or ethnicity			
Alaska Native or American Indian	<10 *	<10 *	<10 *
Asian or Pacific Islander	<10 *	<10 *	<10 *
Hispanic or Latina	25 (4%)	10 (6%)	35 (5%)
Non-Hispanic Black or African American	67 (12%)	26 (14%)	93 (12%)
White	477 (83)	139 (77%)	616 (81)
Unknown	<10	<10	<10
Education			
Less than high school	37 (6%)	15 (8%)	41 (5%)
High school or GED	120 (21%)	43 (24%)	166 (22%)
School after high school	209 (36%)	74 (41%)	284 (38%)
College degree or higher	207 (36%)	48 (27%)	267 (35%)
Unknown	<10	<10	0
Income			
<$20,000	104 (18%)	56 (31%)	165 (22%)
$20,000–$35,000	171 (30%)	47 (26%)	232 (31%)
$35,000–$50,000	100 (17%)	37 (20%)	132 (17%)
$50,000–$75,000	85 (15%)	21 (12%)	134 (18%)
>$75,000	81 (14%)	13 (7%)	95 (13%)
Unknown	36 (6%)	<10	0
*Lifestyle factors*			
Alcohol intake (svgs/wk)	0.42; [0, 1.81] *	0.21; [0, 2.73] *	0.21; [0, 1.37]
Total calcium (mg/d)	1024.1; [647.5, 1557.2] *	1029.3; [711.8, 1532.0] *	973.8; [649.7, 1530.9] *
Total folate (mcg/d)	635.47; [420.94, 881.23] *	619.21; [448.25, 861.41] *	593.64; [419.06, 838.82] *
Red or processed meat (svgs/d)	1.87; [1.04, 2.95] *	1.85; [1.1, 3.31] *	1.83; [1.07, 3] *
Energy expenditure (MET-hours/wk)	7.08; [1.5, 16.75] *	7.25; [1.38, 16.42] *	7; [1.5, 15.5] *
History of smoking (current yes/no)	45 (7.84%) *	16 (8.94%)*	47 (6.2%)
Any supplement use	257 (44.54%) *	71 (39.23%)	328 (43.27%)
Uses anti-diabetes medication	24 (4.16%)	16 (8.84%)	35 (4.62%)
Uses anti-hypertensive medication	185 (32.06%)	68 (37.57%)	231 (30.47%)
Uses anti-lipid medication	42 (7.28%)	16 (8.84%)	50 (6.6%)
Uses NSAIDs	196 (33.97%)	58 (32.04%)	262 (34.56%)
*Clinical risk factors*			
Gail 5-year risk score	1.9; [1.16, 2.22]	1.7; [1.1, 1.92]	1.68; [1.15, 1.95]
Family history of cancer			
Yes	80 (14%)	30 (17%)	124 (16%)
No	453 (79%)	139 (77%)	617 (8%)
Unknown	44 (8%)	12 (7%)	17 (2%)
Personal history of cancer			
Yes	27 (5%)	13 (7%)	39 (5.2%)
No	547 (95%)	168 (93%)	719 (95%)
Unknown	<10	0	0
History of colonoscopy			
Yes	253 (44%)	71 (39%)	355 (47%)
No	249 (43%)	90 (50%)	401 (53%)
Unknown	75 (13%)	20 (11%)	<10
History of colon polyp removal			
Yes	39 (7%)	11 (6%)	73 (10%)
No	454 (79%)	145 (80%)	677 (89%)
Unknown	84 (15%)	25 (14%)	<10
History of treated diabetes			
Yes	27 (5%)	19 (11%)	40 (5%)
No	549 (95%)	162 (90%)	717 (95%)
Unknown	<10	0	<10
History of treated hypertension			
Yes	144 (25%)	55 (30%)	186 (25%)
No	356 (62%)	104 (57%)	568 (75%)
Unknown	77 (13%)	22 (12%)	<10
Had at least one term pregnancy	510 (88%)	158 (87%)	676 (89%)
Post-menopausal hormone therapy use			
Never	286 (50%)	97 (54%)	331 (44%)
Past	75 (13%)	30 (17%)	125 (16%)
Current estrogen alone	117 (20%)	36 (20%)	189 (25%)
Current estrogen and progestin	98 (17%)	18 (10%)	113 (15%)
*Study variables*			
WHI enrollment date			
Baseline	108 (19%)	44 (24%)	152 (20%)
Year 1	220 (38%)	65 (36%)	285 (38%)
Year 3	243 (42%)	68 (38%)	311 (41%)
Year 6	<10	<10	<10
Year 9	0	<10	<10
Calcium / Vitamin D (CaD) trial arm			
Not randomized to CaD	462 (80%)	147 (81%)	560 (74%)
Control arm	54 (9%)	19 (11%)	104 (14%)
Intervention arm	61 (11%)	15 (8%)	94 (12%)
Dietary Modification (DM) trial arm			
Not randomized to DM	384 (67%)	122 (67%)	488 (64%)
Control arm	118 (20%)	27 (15%)	163 (22%)
Intervention arm	75 (13%)	32 (18%)	107 (14%)
Hormone therapy (HT) trial arm			
Not randomized to HT	495 (86%)	144 (80%)	641 (85%)
Estrogen-only control arm	19 (3%)	13 (7%)	27 (4%)
Estrogen-only intervention arm	15 (3%)	11 (6%)	34 (4%)
Estrogen and progestin control arm	16 (3%)	<10	30 (4%)
Estrogen and progestin intervention arm	32 (6%)	<10	26 (3%)

^1^ For continuous variables, the summaries displayed are: median; inter-quartile range. For binary and categorical variables, the summaries displayed are: count (proportion). * Denotes variables with a nonzero proportion of missing data. Overall, the proportion of missing data ranged from 0% to 11%, with most variables having less than 3% missing data. MET: metabolic equivalent hours per week of recreational physical activity; NSAIDs: non-steroidal anti-inflammatory drugs.

**Table 2 metabolites-14-00463-t002:** Metabolites selected with proportion of explained variation for predicting breast cancer and colorectal cancer ^1^.

Metabolites Selected	Proportion of Explained Variation ^2^	Direction of Coefficient for Metabolites ^3^
**Breast Cancer**	All covariates + metabolites: 0.27	
*Serum:*		
*LC-MS*		
Azelaic acid	0.23	−
Choline	0.23	+
Cysteinyl glycine	0.23	−
Ethanolamine	0.23	+
Gamma tocopherol	0.23	+
Hippuric acid	0.23	−
Isovaleryl carnitine	0.23	+
*N*-isovaleryl glycine	0.23	−
Sucrose	0.23	−
Trimethylamine-*N*-oxide	0.23	+
Valine	0.23	+
Xylose	0.23	−
*Lipidyzer* ^4^		
Cholesteryl ester (CE 12:0)	0.23	−
Cholesteryl ester (CE 20:0)	0.23	−
Diacylglycerol (DAG 14:1)	0.24	+
Free fatty Acid (FFA 18:4)	0.23	−
Free fatty Acid (FFA 20:2)	0.23	+
Hexosylceramide (HCER 22:0)	0.23	+
Hexosylceramide (HCER 22:0)	0.23	−
Phosphatidylcholine (PC 18:1)	0.23	+
Phosphatidylcholine (PC 18:2)	0.23	+
Phosphatidylcholine (PC 16:0/18:2)	0.24	−
Phosphatidylethanolamine (PE 18:2)	0.23	+
Triacylglycerol (TAG 12:0)	0.23	−
Triacylglycerol (TAG 16:0)	0.23	−
Triacylglycerol (TAG 18:0)	0.23	−
Triacylglycerol (TAG 47:0/15:0)	0.23	−
Triacylglycerol (TAG 48:4/18:2)	0.23	−
Triacylglycerol (TAG 50:0/16:0)	0.23	+
Triacylglycerol (TAG 50:2/18:2)	0.23	−
Triacylglycerol (TAG 50:5/18:3)	0.24	−
Triacylglycerol (TAG 52:2/18:2)	0.24	+
Triacylglycerol (TAG 55:4/18:1)	0.23	−
*Urine*		
*NMR*		
Dimethylamine	0.23	−
Propanediol	0.23	−
Formate	0.23	+
Sucrose	0.23	−
Taurine	0.23	+
Uracil	0.23	−
Trimethylamine-*N*-oxide	0.23	−
2-Hydroxyisobutyrate	0.23	+
2-Oxoglutarate	0.23	−
*GC-MS*		
Unknown 73.012.10 ^5^	0.23	−
Unknown 73.014.49 ^5^	0.23	+
Unknown 73.016.52 ^5^	0.23	+
Colorectal Cancer	All covariates + metabolites: 0.31	
*Serum*		
*LC-MS*		
Adenosine	0.23	−
Leucic Acid	0.21	+
Glycerate	0.25	+
Myo-inositol	0.22	+
N-Acetyl-glutamate	0.22	−
N-Acetyl-glycine	0.23	+
N-Acetylneuraminate	0.22	+
2-Hydroxyglutarate	0.22	+
Hydroxyproline	0.21	+
7-Methylguanine	0.22	+
*Lipidyzer* ^4^		
Lysophosphatidylcholine (LPC 20:3)	0.22	−
*Urine*		
*NMR*		
Acetate	0.21	+
Allantoin	0.21	−
Histidine	0.22	−
Isoleucine	0.21	+
Taurine	0.22	+
Threonine	0.21	+
Trimethylamine-*N*-oxide	0.21	+
Uracil	0.22	−
*GC-MS*		
Unknown 103 17.03 ^5^	0.21	−
Unknown 285 22.41 ^5^	0.22	+
Unknown 57 9.58 ^5^	0.22	+
Unknown 73 10.76 ^5^	0.21	−
Unknown 73 17.66 ^5^	0.21	+

^1^ All variables listed below were selected by either the lasso or SL selection procedure in the corresponding platform-specific analysis. The base set of covariates (forced into all models) were age, WHI enrollment date, and self-reported race or ethnicity. Selected covariates for breast cancer: education level, income, alcohol intake, current smoking, total folate intake, Gail 5-year risk, family history of CRC, prior removal of ≤1 colon polyp, currently using estrogen, waist circumference, BMI (kg/m^2^), randomized to CaD or HT, date of sample draw visit. Selected covariates for colorectal cancer: age, self-reported race/ethnicity, education, income, alcohol intake, total folate intake, waist circumference, BMI (kg/m^2^), ≥1 colonoscopy, prior removal of ≥1 colon polyp, sample draw visit, randomized to DM control arm. ^2^ The proportion of explained variation (PEV) was estimated by first creating a dataset with only the selected metabolites and covariates for each outcome. Then, we used cross-validation to fit a logistic regression on each set of training data and predict on the test data; the PEV is defined as the correlation between the observed outcomes and the predictions. ^3^ Positive direction of the estimated coefficient from the multiple logistic regression model implies higher odds of being a case; negative direction implies lower odds of being a case. ^4^ In CE, X:A; FFA, X:A; DAG, X:A/Y:B; HCER, X:A; PC, X:A/Y:B; PE, X:A/Y:B; and LPC, X:A, X and Y indicate the number of carbon atoms and A and B indicate the number of double bonds in the fatty acid chains. Lipids without both A and B represent the sum of all fatty acids in that class. For example, DAG (14:1) equals the sum of all diacylglycerol, i.e., summing all DAG (x/14:1) and DAG (14:1/x). ^5^ Values represent mass at retention time of the unknown metabolites, i.e., 73 12.10 indicates a mass of 73 at 12.10 min. In TAG, X:A/Y:B, X indicates the total number of carbon atoms and A indicates the total number of double bonds in the three fatty acid chains, and Y indicates the number of carbon atoms and B indicates the number of double bonds in one of the fatty acid chains.

**Table 3 metabolites-14-00463-t003:** Metabolite predictors of colorectal cancer derived across all platforms and prediction algorithms, and sensitivity analyses, along with class and function ^1^.

Metabolite	Class	Function/Relevance
2-Hydroxyglutarate	Hydroxy acid	TCA intermediate; inhibitor of alpha-keto dehydrogenases, including histone demethylases; considered an oncometabolite
*N*-Acetyl-glycine	Alpha amino acid	Lipid signaling mediator
Taurine	Sulfur-containing amino acid	Metabolism of fats, bile acids
Threonine	Amino acid	Metabolism of fats
TAG (53:2/FA18:1)	Triglyceride	Fat storage; energy
LPC (FA20:3)	Lysophosphatidyl choline	Cholesterol metabolism
CE (FA20)	Cholesteryl ester	Cholesterol metabolism
Acetate	Short chain fatty acid	Microbial metabolite; acetylation reactions, energy metabolism
Glycerate	Sugar acid	Generation of ATP
Adenosine	Nucleoside	Energy transfer, component of RNA/DNA
Hypoxanthine	Purine	Metabolism of adenosine
Uracil	Nucleic acid	Component of RNA
7-Methylguanine	Purine	Component of RNA/DNA; potential biomarker of chicken
Histidine	Amino acid	Protein synthesis, histamine and carnosine biosynthesis, scavenger of ROS
Leucic acid	Hydroxy fatty acid	Leucine (branched-chain amino acid) metabolite; accelerates lipid peroxidation, oxidative stress
Isoleucine	Branched-chain amino acid	Protein metabolism, hemoglobin production, glucose control, immunity
*N*-Acetyl-glutamate	Alpha amino acid	Involved in the urea cycle
Allantoin	Imidazoles	Microbial metabolite; uric acid metabolite; found in dairy
*N*-Acetyl-neuraminate	Amino sugar	Component of glycoproteins and mucins involved in immunity
Hydroxyproline	Amino acid	Component of collagen; biomarker of meat
Myo-inositol	Sugar alcohol	Component of phosphatidylinositol; increases insulin sensitivity; biomarker of whole grains
Trimethylamine-*N*-oxide	Amine oxide	Microbial metabolite formed from choline, betaine, and carnitine; associated with cardiovascular disease; biomarker of fish and red meat

^1^ Class and function ascertained from PubChem or Human Metabolome Database; unnamed metabolites not included. TAG: triacylglyceride; LPC: lysophosphatidyl choline; CE: cholesterol ester.

**Table 4 metabolites-14-00463-t004:** Metabolites selected for predicting breast cancer and colorectal cancer in pooled analysis ^1^.

Metabolites Selected	Proportion of Explained Variation ^2^	Direction of Coefficient for Metabolites ^3^
**Breast Cancer**	All covariates + metabolites: 0.27	
*Serum*		
*LC-MS*		
Cystenyl-glycine	0.22	−
Ethanolamine	0.21	+
Sucrose	0.22	−
*Lipidyzer* ^4^		
Free fatty acid (FFA 20:2)	0.22	+
Phosphatidylcholine (PC 16:0/18:2)	0.23	+
Triacylglyceride (TAG 48:4/18:2)	0.22	−
Triacylglyceride (TAG 50:5/18:3)	0.22	−
Triacylglyceride (TAG 52:2/18:2)	0.22	+
*Urine*		
*NMR*		
Uracil	0.22	−
2-Hydroxyisobutyrate	0.22	+
*GC-MS*		
Unknown 73.0 14.49 ^5^	0.21	+
Unknown 73.0 12.10 ^5^	0.22	−
**Colorectal Cancer**	All covariates + metabolites: 0.33	
*Serum*		
*LC-MS*		
Adenosine	0.20	−
Leucic acid	0.18	+
Glycerate	0.23	+
Hypoxanthine	0.18	+
Myoinositol	0.19	+
*N*-Acetylneuraminate	0.19	−
2-Hydroxyglutarate	0.19	+
7-Methylguanine	0.19	+
*Lipidyzer*		
CE (FA20)	0.17	−
TAG (53:2/18:1)	0.18	+
*Urine*		
*NMR*		
Histidine	0.18	−
Taurine	0.19	+
Threonine	0.17	+
*GC-MS*		
Unknown 103 17.03 ^5^	0.18	−
Unknown 285 22.41 ^5^	0.18	+
Unknown 57 9.58 ^5^	0.18	+
Unknown 73 10.76 ^5^	0.18	−
Unknown 73 17.27 ^5^	0.17	+

^1^ All variables listed below were selected using the lasso for variable selection with all four platforms pooled together prior to variable selection. The base set of covariates (forced into all models) are age, WHI enrollment date, and self-reported race or ethnicity. Selected covariates for breast cancer: age, self-reported race/ethnicity, income, Gail 5-year risk score, waist circumference, sample draw visit, randomized to the CaD control arm. Selected covariates for colorectal cancer: age, self-reported race/ethnicity, income, education, waist circumference, sample draw visit. ^2^ The proportion of explained variation (PEV) was estimated by first creating a dataset with only the selected metabolites and covariates for each outcome. Then, we used cross-validation to fit a logistic regression on each set of training data and predict on the test data; the PEV is defined as the correlation between the observed outcomes and the predictions. ^3^ Positive direction of the estimated coefficient from the multiple logistic regression model implies higher odds of being a case; negative direction implies lower odds of being a case. ^4^ FFA: free fatty acid; FA: fatty acid; TAG: triacylglyceride; PC: phosphatidyl choline; CE: cholesterol ester. ^5^ Values represent mass at retention time of the unknown metabolites, i.e., 73 12.10 indicates a mass of 73 at 12.10 min. In post hoc analyses excluding women using HT, prediction performance in the subpopulation was modestly improved for CRC compared to the full population (CV-AUCs range from 0.622–0.637, while in the full population they range from 0.589–0.608). Prediction performance for BC was slightly decreased in the subpopulation compared to the full population (CV-AUCs range from 0.535–0.554, while in the full population they range from 0.559–0.563). Several LC-MS metabolites were selected in the subgroup analysis that were also selected in the whole-cohort analysis: cysteinyl glycine, *N*-isovaleryl glycine, and valine (BC); adenosine, leucic acid, glycerate, hydroxyproline, and 2-hydroxyglutarate (CRC). Additional metabolites selected in the subgroup analysis for CRC included adipic and 3-hydroxybutyric acids, involved in fatty acid metabolism; betaine, a marker of whole grains; glucuronate, found in gums and fermented beverages; and trigonelline, found in coffee.

## Data Availability

Data, codebook, analytic code used in this report may be accessed in a collaborative mode as described on the Women’s Health Initiative website (www.whi.org).

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
