# Peer review of "Metabolite Predictors of Breast and Colorectal Cancer Risk in the Women’s Health Initiative"

_metabolites, 2024, doi:10.3390/metabo14080463_

Round 1

Reviewer 1 Report

Comments and Suggestions for Authors

The manuscript under review is a prospective study devoted to the search of breast cancer and colorectal cancer biomarkers followed by development of statistical models. The importance of the study is that the authors used serum and urine samples collected years before the manifestation of the diseases.

Having considered the manuscript, I have following comments.

1) Table 1: I have a question whether it is possible and necessary to add the data on the use of contraceptives by the participants of the study when they were young women. Can this factor contribute to development of BC? There are articles in the literature that oral contraceptives have little or no influence in women except when they are used in young age. As such, there is a possibility of the presence of some bias caused with the use of the contraceptives by the study participants.

2) Section 2.3.1: Please provide al details on serum metabolites and lipids analysis (sample preparation protocol, list of the metabolites/lipids screened, LC and MS/MS parameters used for separation and detection of the compounds). IN addition, I suggest authors to provide raw data obtained after metabolites screening so that other researchers could used it for checking and/or developing other statistics models.

3) Page 13, Table 4: Please correct 7-Methylguanine (dash is necessary).

4) Page 14, Table 4 (the end, GC-MS data): Please add an explanation of the notification of the unknown compounds both in the footnote and in the text. What do the numbers stand for?

5) Sections 3 and 4: In addition to the search of biomarkers, I suggest authors to perform a minimal analysis of metabolic pathways disturbed at the initial stages of cancer development. Moreover, I recommend comparing the results of this analysis with the literature data on the same topic.

6) Page 16, line 481: A misprint is here (serum phosphotyidylcholines).

7) Page 18, Data availability statement: In additional to my comment above: please add a reference number or dataset ID to access the data related to the study presented.

Author Response

Reviewer #1:

The manuscript under review is a prospective study devoted to the search of breast cancer and colorectal cancer biomarkers followed by development of statistical models. The importance of the study is that the authors used serum and urine samples collected years before the manifestation of the diseases.

Having considered the manuscript, I have following comments.

1) Table 1: I have a question whether it is possible and necessary to add the data on the use of contraceptives by the participants of the study when they were young women. Can this factor contribute to development of BC? There are articles in the literature that oral contraceptives have little or no influence in women except when they are used in young age. As such, there is a possibility of the presence of some bias caused with the use of the contraceptives by the study participants.

RESPONSE: We thank the reviewer for the comment. While there have been studies showing an association between contraception use and breast cancer risk, the risk depends on recency of use, with studies showing risk with current or recent use in pre-menopausal women, but no excess risk after 10 years of discontinuing use, or among post-menopausal women.[1-4] Our women were all post-menopausal at enrollment. More importantly however, our aim was to determine whether the addition of blood and urine metabolites performed better than well-known risk factors. We did not see any improvement in the risk prediction with addition of metabolites compared to the risk factors alone. In fact, the largest CV-AUC for models including metabolites in addition to risk factors was less than that for risk factors alone. Therefore, addition of this covariate would not change our outcome with regard to the metabolites.

2) Section 2.3.1: Please provide al details on serum metabolites and lipids analysis (sample preparation protocol, list of the metabolites/lipids screened, LC and MS/MS parameters used for separation and detection of the compounds). IN addition, I suggest authors to provide raw data obtained after metabolites screening so that other researchers could used it for checking and/or developing other statistics models.  

RESPONSE: Thank you for the suggestion. We have expanded the sections for metabolite methods with the information suggested. Further, we have provided supplemental tables listing all metabolites targeted and transitions on each platform (see Supplemental Table 4).

Regarding provision of raw metabolite data, similarly to other large scale prospective studies, i.e., Framingham Heart Study, data are not uploaded to data repositories. WHI policy requires that data be requested in a collaborative manner as indicated in our data sharing statement, i.e., the data, codebook, and analytic code can be requested as described on the Women’s Health Initiative website (www.whi.org).

3) Page 13, Table 4: Please correct 7-Methylguanine (dash is necessary).

RESPONSE: We have added a dash to this metabolite.

4) Page 14, Table 4 (the end, GC-MS data): Please add an explanation of the notification of the unknown compounds both in the footnote and in the text. What do the numbers stand for?

RESPONSE: We have added a footnote indicating that these values represent the mass at retention time. Thus 73 12.10 represents a mass of 73 daltons at a retention time of 12.10 minutes.

5) Sections 3 and 4: In addition to the search of biomarkers, I suggest authors to perform a minimal analysis of metabolic pathways disturbed at the initial stages of cancer development. Moreover, I recommend comparing the results of this analysis with the literature data on the same topic. 

RESPONSE: We thank the reviewer for this comment. We did attempt to perform pathway analyses. However, our efforts were hampered by the fact that, 1) the majority of lipids are not included in pathway modules; 2) the majority of GC-MS metabolites are derivatives and are also not contained in most databases. Thus, we were left with LC-MS and NMR metabolites (only ~200 of the >1,000 metabolites) which provided very sparse pathway coverage with 1-3 metabolites at most within any given pathway. We therefore, provide a table of the metabolites that were predictive of CRC (there were not any metabolites that provided prediction performance over risk variables), and their functions.

6) Page 16, line 481: A misprint is here (serum phosphotyidylcholines).

RESPONSE: We thank the reviewer for catching that typo and have corrected the spelling.

7) Page 18, Data availability statement: In additional to my comment above: please add a reference number or dataset ID to access the data related to the study presented.

RESPONSE:  This is WHI ancillary study 498 (R01 CA119171)

  1. Collaborative Group on Hormonal Factors in Breast, C., Breast cancer and hormonal contraceptives: collaborative reanalysis of individual data on 53 297 women with breast cancer and 100 239 women without breast cancer from 54 epidemiological studies. Lancet 1996, 347 (9017), 1713-27.
  2. Satish, S.; Moore, J. F.; Littlefield, J. M.; Bishop, I. J.; Rojas, K. E., Re-Evaluating the Association Between Hormonal Contraception and Breast Cancer Risk. Breast Cancer (Dove Med Press) 2023, 15, 227-235.
  3. Dumeaux, V.; Fournier, A.; Lund, E.; Clavel-Chapelon, F., Previous oral contraceptive use and breast cancer risk according to hormone replacement therapy use among postmenopausal women. Cancer Causes Control 2005, 16 (5), 537-44.
  4. Kanadys, W.; Baranska, A.; Malm, M.; Blaszczuk, A.; Polz-Dacewicz, M.; Janiszewska, M.; Jedrych, M., Use of Oral Contraceptives as a Potential Risk Factor for Breast Cancer: A Systematic Review and Meta-Analysis of Case-Control Studies Up to 2010. Int J Environ Res Public Health 2021, 18 (9).

Reviewer 2 Report

Comments and Suggestions for Authors

1. In Table 1, why is information about the percentage of missing data needed if it is almost always zero? It seems to me that this information weighs down the table.

2. The authors provide the mean and interquartile range, but in this case it is necessary to provide the median and interquartile range or the mean and standard deviation. I would still recommend providing the median and range in the usual form as a 25-75% interval (the first and third quartiles).

3. Why is information about Lifestyle factors, Clinical risk factors, Education and Income provided? Is this information used in creating a prognosis model for breast and colorectal cancer? It seems that this part of the work stands alone, and the rest of the article is separate. However, a combined model could provide more accurate prognosis results. For example, patients with a family history of cancer may have potentially prognostically important metabolites that are different from those that are important for the subgroup without a family history of cancer. This is just one possible assumption.

Author Response

Reviewer #2:

  1. In Table 1, why is information about the percentage of missing data needed if it is almost always zero? It seems to me that this information weighs down the table.

RESPONSE: Thank you for this suggestion. We have removed the % missing data and added the information as a footnote.

  1. The authors provide the mean and interquartile range, but in this case it is necessary to provide the median and interquartile range or the mean and standard deviation. I would still recommend providing the median and range in the usual form as a 25-75% interval (the first and third quartiles).

RESPONSE: Thank you for this suggestion, we have altered our table accordingly.

  1. Why is information about Lifestyle factors, Clinical risk factors, Education and Income provided? Is this information used in creating a prognosis model for breast and colorectal cancer? It seems that this part of the work stands alone, and the rest of the article is separate. However, a combined model could provide more accurate prognosis results. For example, patients with a family history of cancer may have potentially prognostically important metabolites that are different from those that are important for the subgroup without a family history of cancer. This is just one possible assumption..

RESPONSE: Thank you for the opportunity to clarify our analysis. Our aim was to determine whether prediagnostic metabolites provided predictive power above and beyond well-established risk factors for these cancers. We conducted a base model, in which we only included the meta-data, including clinical and lifestyle factors, demographics, etc. We then combined those variables with the metabolites and evaluated the performance. We include this information in the final paragraph in the introduction and the first paragraph of the discussion.

Round 2

Reviewer 1 Report

Comments and Suggestions for Authors

As far as I could see, the authors have revised the manuscript in accordance with all my comments and provided rebuttal where necessary. I have no further comments.